# Management of Squamous Cell Carcinoma in Chronic Osteomyelitis: Our Experience, Review of the Literature and Role of MRI in Differential Diagnosis

**DOI:** 10.3390/diagnostics12092062

**Published:** 2022-08-25

**Authors:** Roberto Scanferla, Giuliana Roselli, Guido Scoccianti, Marco Bartolini, Francesco Muratori, Domenico Andrea Campanacci

**Affiliations:** 1Department of Orthopaedic Oncology and Reconstructive Surgery, AOU Careggi, 50139 Florence, Italy; 2Department of Radiology, AOU Careggi, 50139 Florence, Italy

**Keywords:** squamous cell carcinoma, chronic osteomyelitis, MRI, limb salvage, amputation

## Abstract

Background. The authors describe a series of patients with chronic osteomyelitis associated with squamous cell carcinoma, whilst analyzing its incidence in chronic osteomyelitis, surgical options for treatment and focusing on the role of MRI in differential diagnosis. Methods. The authors reviewed 73 cases of chronic osteomyelitis (CO) treated in their department between 1995 and 2019. Six of these patients (8.2%) had a malignant degeneration in squamous cell carcinoma (SCC). All cases with malignancy were evaluated with preoperative gadolinium-enhanced MRI. Results. In this series, the authors observed an incidence rate of 8.2% (6 cases out of 73); all patients were male with a mean age of 63.5 years. The mean time between the occurrence of chronic osteomyelitis and the diagnosis of squamous cell carcinoma was 36 years (range: 21–66). The treatment consisted of amputation in five patients and limb salvage with vascularized fibula autograft in one case. Conclusions. MRI played a key role in the differential diagnosis between infected and tumoral tissue, and was found to be crucial during follow-up. Amputation is the treatment of choice in extended tumoral involvement of bone and soft tissue.

## 1. Introduction

Chronic osteomyelitis represents an inflammatory process associated with tissue necrosis [1] caused by a bacterial infection (usually *Staphylococcal*) involving bone, where host defensive mechanisms and antibiotics are unable to reach the established organisms [2]. During chronicization of the infection, an important role is played by the polysaccharide/protein matrix produced by the microorganism called biofilm. Once the biofilm has matured, bacteria convert themselves to a lowered metabolic rate that reduces their susceptibility to antibiotics [3]. There is a high level of evidence which shows that chronic osteomyelitis promotes carcinogenesis [4,5], particularly with the onset of squamous cell carcinoma [6] of the skin orifice or the fistula tract. Malignant degeneration is a rare and late complication, developing 20 to 40 years after infection onset and involving approximately 1% of these patients [6]. In squamous cell carcinoma arising from chronic osteomyelitis sites, early diagnosis and adequate, aggressive management are critical to the prognosis and final outcome of these patients.

The authors report their experience on the treatment of six cases of osteomyelitis complicated by squamous cell carcinoma, with the aim to answer the following questions: (a) What is the incidence of squamous cell carcinoma in chronic osteomyelitis? (b) Is MRI the optimal imaging technique for differential diagnosis between osteomyelitis and carcinoma? (c) Limb salvage or amputation: which is the appropriate surgical approach?

## 2. Materials and Methods

The authors retrospectively reviewed 73 cases of chronic osteomyelitis (CO) treated in their Department between 1995 and 2019. CO was defined as a long-lasting condition associated with clinical and/or radiographic evidence of bone infection, persisting after adequate and appropriate therapy [2].

Six of these patients (8.2%) had a malignant degeneration in squamous cell carcinoma (SCC). The mean age of patients at tumoral diagnosis was 63.5 years (range: 47–84) and all of them were male. Three of these patients developed CO after an open fracture of the tibia, two after acquiring a skin lesion of the hand involving the bone, and one after receiving corrective surgery for congenital clubfoot. The mean age of patients with trauma was 27.2 years (range: 11–48). The three open-tibial fractures were treated with temporary external fixation followed by ORIF (open reduction and internal fixation), whilst the two deep wounds involving the hand were treated with extensive debridement and the congenital clubfoot was treated with tendon release and deformity correction. After the diagnosis of CO, the treatment was surgical in all cases and consisted of hardware removal and aggressive debridement. Patients underwent an average of 3 surgical procedures (range: 1–9) before the diagnosis of malignant transformation. Microbiologic cultures were positive for *Pseudomonas aeruginosa* and Proteus Mirabilis in one case; for *Pseudomonas aeruginosa* and *Enterococcus fecalis* in another and *Staphilococcus epidermidis* in two patients. *Pseudomonas aeruginosa* and *S. aureus* were assessed in the final two cases.

All cases with malignant degeneration were evaluated with preoperative radiography, CT scan and gadolinium-enhanced MRI. MR examinations were performed using a 1.5 Tesla MRI scanner (AERA, Siemens Medical Solutions, Erlangen, Germany). Unenhanced examination included a coronal and sagittal T1-weighted turbo spin-echo sequence (TSE), a coronal fat suppressed intermediate-weighted spin echo sequence, a transverse T1-w TSE sequence and a transverse T2-w TSE sequence. Dynamic contrast-enhanced MR imaging (DCE-MRI) was performed after the intravenous administration of a gadolinium chelate (0.2 mmol/kg body weight) using an automated power injector at a flow rate of 1.5 mL/s, followed by 20–30 mL of saline at the same flow rate.

A time-resolved magnetic resonance angiography (MRA) with an interleaved stochastic trajectories (TWIST) sequence was used for dynamic images. For each time-resolved MRA sequence, 31 dynamic-phase scanning images with a temporal resolution of 10 s were obtained for covering arterial and venous phases. The total scanning time for dynamic gadolinium enhancement was three minutes. Then, a static coronal T1-w TSE and transverse T1-w with fat saturation were performed.

## 3. Results

The mean time between the occurrence of CO and the diagnosis of SCC was 36 years (range: 21–66) and the malignancy was confirmed with histopathological examination in all cases. None of the patients had distant metastasis at diagnosis.

Plain radiographs showed bone remodeling, marked periosteal thickening, inhomogeneous osteosclerosis and a focal radiolucent lesion with ill-defined margins (Figure 1). Serial radiographs detected changes in cortical bone and periosteal reaction that suggested active infection or malignant transformation. No soft tissue mass was detected on the plain radiograph.

MRI showed abnormal bone marrow signals, the remodeling of cortical bone and periosteal thickening consistent with chronic osteomyelitis (Figure 2). Cortical disruption corresponding to the sinus tract was well observed on the non-contrast MRI, as was the soft tissue involvement with the tumoral and inflammatory process. Contrast medium was useful for the better characterization of soft tissue changes, allowing the assessment of tumoral extension, and differentiating active infection from malignant transformation (Figure 3 and Figure 4).

The primary surgical treatment consisted of amputation in four patients: of the second and third ray of the hand in one patient, of the second finger of the hand in the second patient, above-knee amputation in the third, and below-knee in the final patient. The patient who underwent a below-knee amputation was amputated after an inadequate debridement of a lesion of the leg (Figure 3 and Figure 4). Another patient with a small lesion of the foot underwent an excision with wide margins. Two of these patients had a local recurrence, which was treated with a below-elbow amputation in one case and an above-knee amputation in the other. Lastly, one patient underwent a limb salvage procedure with intercalary resection of the affected bone and two-stage reconstruction with the osteoinductive membrane technique. A non-union occurred, requiring surgical revision and reconstruction with a vascularized fibula autograft. At latest follow-up, a complete fusion of the fibular graft was evident.

All patients during surgery underwent a sentinel lymph node biopsy and none had limphnodal metastasis. Moreover, none of the patients underwent radiation- or chemotherapy after or before surgery.

The mean clinical follow-up (FU) time was 74 months (range: 27–132). Four patients were CDF (continuously disease free) and two were disease free after recurrence treatment.

## 4. Discussion

Carcinomatous degeneration associated with chronic osteomyelitis is a rare complication that occurs after 20–40 years, generally involving the path of the fistula or the orifice of the skin [7,8,9]. It generally affects long bone infections and the most frequent histotype is squamous cell carcinoma, or more rarely, sarcoma or lymphoma [5,8,10]. The physio-pathological mechanism of this degeneration is debated and not well known. Chronic irritation of the skin or the exposure of the soft tissue to different growth factors play an important role [5,11]. Under these conditions, the immune system induces the production of mediators and cytokines that would modulate the gene expression of various proteins responsible for carcinogenesis processes, including p53 [11,12,13,14,15]. Furthermore, gene mutation and horizontal gene transfer have been found to be characteristic of lesions with multiple bacterial infections, which may contribute to the carcinogenesis process [13]. Factors such as ultraviolet radiation, epithelial hyperplasia and long-standing purulent exudates, abrasion and other mechanical stimuli can exaggerate the malignant transformation [16]. There is evidence that Gram-positive flora can be replaced by predominant Gram-negative flora that produce endotoxins associated with cancer [17].

Hawkins was the first to report a case of carcinomatous degeneration on chronic osteomyelitis in 1835 [18]. The largest series of squamous cell tumors in chronic osteomyelitis were reported by the Mayo Clinic group (with about 23 cases) and by the Rizzoli Institute (with 33 cases) [19,20].

The incidence of malignant transformation is estimated at between 0.2% and 1.7% for patients with osteomyelitis, although Johnston et al. [9] reported much higher proportions, 23% out of a total of 4000 cases studied. In developing countries, the incidence is higher due to delayed diagnosis and inadequate treatment. Traumatic injuries remain the most frequent causes of osteomyelitis [7], with an infection rate ranging from 4% to 16% in long bone open fractures, while hematogenous forms have a much lower incidence [7].

Moura et al. [21] reported six cases of chronic osteomyelitis related to malignant transformation, with a diagnosis of squamous cell carcinoma in all patients. Li et al. [22] reported eight cases of squamous cell carcinoma after chronic osteomyelitis. Peng et al. [23] reported 6 cases of malignant transformation out of a total of 192 cases of chronic osteomyelitis, with an average free interval time of 31 years; in all patients, the leg was the affected site. In this series, the authors observed an incidence rate of 8.2% (6 cases out of 73). All patients were male with a mean age at diagnosis of 63.5 years.

The duration of osteomyelitis appears to be the main risk factor related to carcinogenesis, with a latent period of at least 20 years [6,7,8], considering that this complication can occur earlier [24]. Alami reports a prevalence in males, with an average time of malignant transformation of about 24.5 years [13]. Similarly, the authors of this series observed an average transformation time of about 23 years.

The most affected bone segment in this series was the tibia, with three cases (50%). Similar data are also reported in the literature [13,24]. On the contrary, the upper limb is rarely involved, but in this study, the hand was affected in two patients (33%).

Clinical signs that should alert the presence of a malignant transformation include increasing pain, bleeding, malodor, progressive osteolysis and erosion of the bone, and progressively growing soft tissue masses around the fistula or ulceration [7,8,10,13,24,25]. In this series, the presence of fistula was constant, associated with a swelling of the soft tissues. Bone involvement, such as osteolytic lesions, occurred in all six cases.

Histological diagnosis was obtained through a biopsy in all cases, but imaging is crucial for early differential diagnosis between osteomyelitis and malignant transformation. Conventional radiograms usually demonstrate irregular, ill-defined lytic bone lesions, surrounded by sclerosis of various degree and extension, cortical erosions and periosteal thickening [26,27]. A CT scan is an excellent tool for systemic staging and evaluating the extent of bone involvement. It represents the best technique to assess characteristic bone changes of chronic osteomyelitis, including necrotic areas, sequestrum and draining sinuses. A CT scan can also detect soft tissue abscess and air, but it is less sensitive than MRI for detecting chronic alterations related to malignant transformation [26,28]. MRI, because of its high contrast resolution, represents the optimal imaging modality for the assessment of bone changes and soft tissue involvement in chronic osteomyelitis. Necrotic bone sequestrum usually appears dark on all sequences and can be surrounded by T2-weighted fat sat hyperintense and contrast-enhancing hypervascular granulation tissue. Draining sinus tracts are generally visualized as linear or curved T2-weighted hyperintense tracks, crossing from the medullary cavity, through cortical bone and soft tissues, to the skin surface. Sinus tracts may enhance after intravenous contrast administration. Carcinomatous degeneration in chronic osteomyelitis usually arises from the sinus tract and may progress towards the skin surface or infiltrate the surrounding muscle layers. Squamous cell carcinoma has a low signal on MRI. It typically appears isointense to the muscle in T1w sequences and low-to-intermediate in T2w/T2w fat sat sequences, with weak or mild enhancement, homogeneous or heterogeneous, and in a striated pattern after the administration of contrast medium, depending on the tumor differentiation grade and tissue keratinization [29,30] (Table 1). Tumoral tissue may either extend along the sinus tract wall reaching the subcutaneous layers, or infiltrate surrounding muscles. In this case, malignant transformation is difficult to ascertain due to the similarity in signal intensity between the tumor and the surrounding muscle (Figure 3). Conversely, in cases of bone or subcutaneous fat neoplastic infiltration (Figure 2), the diagnosis of malignant degeneration is easier due to the high intrinsic signal difference between tumor and normal tissues. Differential diagnosis is challenging as there is little signal intensity difference between muscle and tumoral tissue on baseline examination. Contrast-enhanced MRI techniques, including magnetic resonance angiography (MRA), increase diagnostic sensitivity and highlight contrast uptake differences between normal and neoplastic tissue [27] (Figure 4). Even though the MRI appearance of squamous cell carcinoma is not specific, being similar to other sarcomatous malignancies that may complicate chronic osteomyelitis, the integration of MRI evidences with clinical and histopathological data may increase diagnostic accuracy. In the authors’ opinion, contrast-enhanced MRI should be performed annually during the follow-up period for patients with CO, particularly in patients with a long history of disease or with clinical evidence of long-lasting symptoms such as: pain, foul smell, exophytic mass, or lack of fistula or ulcer response to adequate therapy. Thru-cut or incisional biopsy is necessary to confirm the diagnosis of SCC, but gadolinium-enhanced MRI enables the early detection of the malignancy and the ability to plan surgery options, i.e., to decide between limb salvage surgery and amputation.

Amputation represents the recommended treatment in the presence of carcinomatous degeneration in chronic osteomyelitis. Indeed, it offers a relatively simple solution for removing both the chronic osteomyelitis and the malignancy [13]. Even if burdened by anxiety, social discomfort and depression resulting from the loss of the limb, amputation provides a faster recovery and it represents an oncologically safer procedure, particularly in locally advanced cases [13]. Especially in the lower limb region, below-knee amputation often guarantees good functional results and quality of life [8,13,31]. Amputation is often required due to the extensive involvement of soft tissue by squamous cell carcinoma and wide skin infiltration. Five of our patients were managed with amputation, as a primary treatment in three cases and after local recurrence in two cases. In patients with localized disease, conservative treatment can be performed. Concerning our series, it was possible in only one case of tibial localization to perform limb salvage surgery with an intercalary reconstruction with an autologous vascularized fibula as a salvage procedure after the failure of the osteo-inductive membrane technique. Conservative treatment is rarely indicated due to the aggressive behavior of squamous cell carcinoma and the extensive infective and tumoral involvement of both bone and soft tissues. When feasible, local excision with wide margins should be achieved and biological reconstructive techniques should be favored, such as osteo-inductive membrane, free vascularized fibula, or bone transportation, thus avoiding prosthesis due to the high risk of infection recurrence.

Some studies reported the use of chemotherapy and/or radiotherapy in multimodal treatment; however, with different protocols and limited information reported, both chemotherapy and radiotherapy were mainly used in patients with local recurrence and/or metastasis and in high-grade tumors [8,24,25,32,33,34,35]. The incidence of local recurrence in two recent reviews ranged between 11% and 16.7%, while the metastasis rate ranged between 8% and 12.0% [24,36], with local lymph nodes and lung as the sites most frequently affected [36]. Most cases of metastasis arise within the first 18 months and patients with no metastatic disease during the first 3 years usually have a good prognosis [35]. After amputation, the incidences of local recurrence, metastasis and disease-related death were reported as 13.0%, 13.5% and 13.5%, respectively [36]. Interestingly, Corrigan et al. [24] reported non-statistically significant differences in disease-free survival and disease recurrence or death at one and five years after amputation, versus wide local excision.

Cancer prognosis is related to multiple factors such as histological type, tumor stage, performance status and treatment strategy. Corrigan et al. [24] reported incidental diagnosis and delayed diagnosis as factors associated with poorer prognosis. Meanwhile, Jiang et al. [36] found that the prognosis of squamous cell carcinoma is not related to the etiology of chronic osteomyelitis, the duration of symptoms, or the extension of bone invasion at diagnosis and treatment strategy; it is probably influenced by histological type in terms of differentiation degree, presence of metastasis and by lymph node involvement at diagnosis. They reported that the incidence of local lymphadenopathy at diagnosis in patients with squamous cell carcinoma was 29%, but only 35% of these cases were characterized as metastasis, while 65% were characterized as reactive inflammatory hyperplasia [36]. It remains controversial whether regional lymphadenectomy should be performed at the time of primary surgery, considering that the increase in size (in most cases) is related to inflammatory lymphadenopathy, resolving spontaneously after surgery [8]. The authors argued that if the increase in size of the lymph nodes persists from 6 to 12 weeks after amputation, their excision is essential [8,12]. In our opinion, a crucial role is played by PET/CT scans and by the biopsy of sentinel lymph nodes, as reported by other authors [25]. In particular, PET/CT is very useful in differential diagnosis between reactive and metastatic lymphadenopathy and the study of sentinel lymph nodes can be recommended, as in other malignancies with a tendency for lymphatic metastatic spread, such as Merkel’s disease or melanoma, especially in clinically node-negative patients.

## 5. Conclusions

The onset of squamous cell carcinoma on osteomyelitis is a rare occurrence; however, when it happens, it often poses problems of differential diagnosis with osteomyelitis itself. The change in the clinical scenario is an alert leading us towards a diagnosis of tumoral transformation. The role of MRI during the follow-up of chronic osteomyelitis is crucial in differential diagnosis between infected and tumoral tissue. Although amputation remains the treatment of choice, an early diagnosis of malignant transformation might improve the chance of conservative treatment, with a positive influence on prognosis.

## Figures and Tables

**Figure 1 diagnostics-12-02062-f001:**
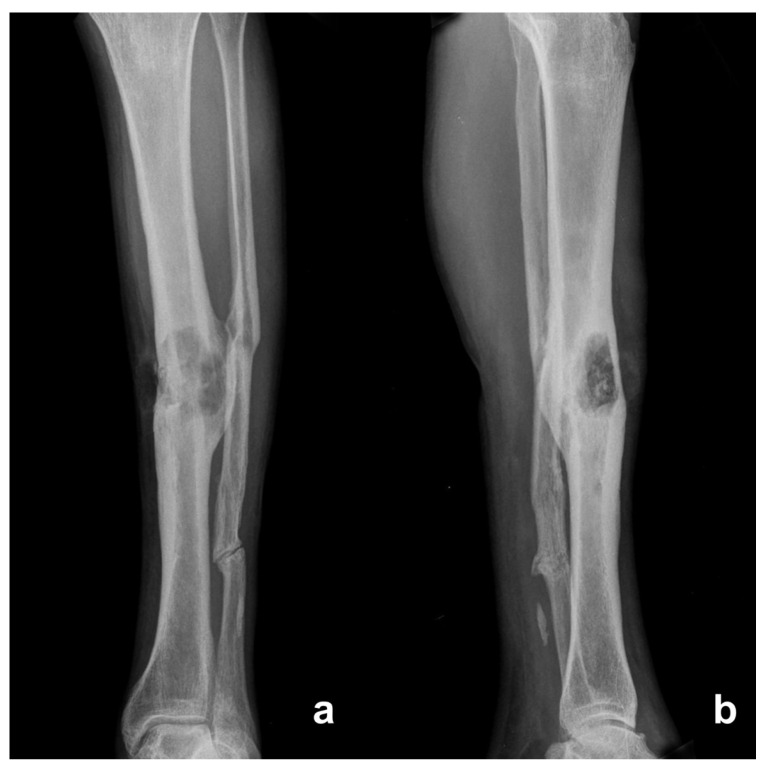
Antero-posterior (**a**) and lateral (**b**) radiographs show ill-defined radiolucent lesions, periosteal thickening disruption of the medial cortex compatible with chronic osteomyelitis and osteolysis of the middle third of the tibial shaft.

**Figure 2 diagnostics-12-02062-f002:**
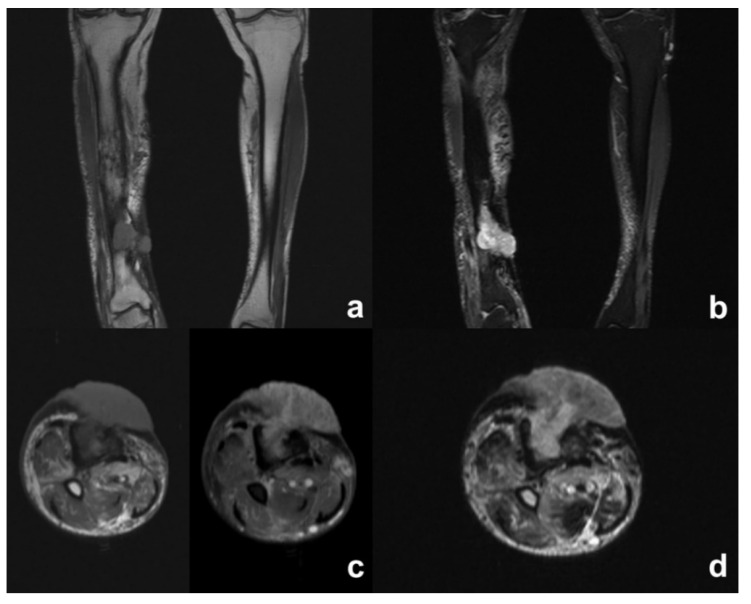
Squamous cell carcinoma of the right leg. (**a**) Cor T1-w shows deformity of the tibial diaphysis with abnormal marrow signal intensity and cortical destruction on the anterior aspect of the distal third of the tibia. There is a large mass developing in soft peridiaphyseal tissue. (**b**) Cor STIR shows intermediate-to-high signal of the mass and mild edema of the subcutaneous tissue of the right leg. (**c**) Tra T1-w pre- and post-medium administration shows a large fungating mass developing from medullary canal beyond the subcutaneous tissue and skin, and post-gadolinium enhancement of the endocanalar tissue. (**d**) Tra T2-w displays an intermediate intensity signal of the tumoral tissue.

**Figure 3 diagnostics-12-02062-f003:**
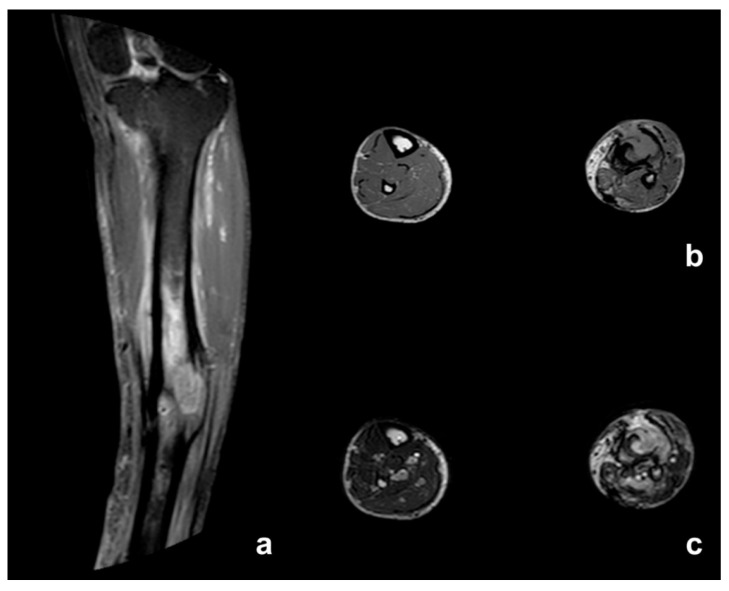
Pre-debridement non-contrast MRI. (**a**) Cor STIR shows periosteal thickening of the left tibial cortex and intramedullary edema around the pseudonodular area of intermediate intensity signal. (**b**) Tra T1w displays periosteal thickening of the posterior cortex of the left tibia and double sinus tract that leads from medullary cavity to surrounding soft tissues. (**c**) Tra T2w shows intramedullary high signals extending along sinus tracts, described as intra-extraosseous abscesses.

**Figure 4 diagnostics-12-02062-f004:**
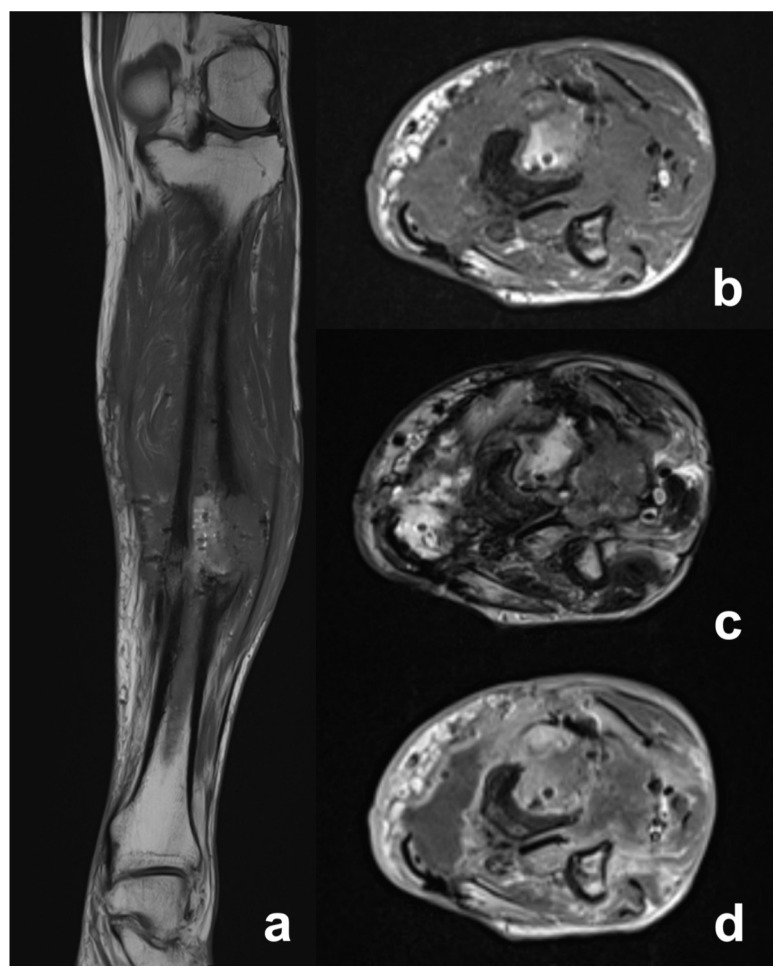
Post-debridement MRI. (**a**) Cor T1-w shows abnormal, low marrow intensity signals of the tibial diaphysis around the area communicating with fistula. On the lateral side of tibial cortex, there is an ill-defined soft tissue mass isointense to muscle. (**b**) Tra T1w shows the same features of the pre-operative magnetic resonance. (**c**) Tra T2 w image demonstrates lateral mass in continuity with the lateral sinus tract and high signal of medullary canal. The mass shows well-defined margins and homogeneous low-intensity signal. It is visible on the medial aspect of the other sinus tract and a heterogeneous fluid collection is developing on the posteromedial side with intermediate-to-high signal. (**d**) Fat-sat Tra T1-w gadolinium-enhanced MR image shows mild enhancement of periosteum, as well as of lateral sinus tract and adjacent mass. On the other aspect, significant enhancement of the medial sinus tract is present, as is a large fluid collection with peripheral enhancement, which characterizes an intramuscular abscess.

**Table 1 diagnostics-12-02062-t001:** General MRI features of acute osteomyelitis (AO), chronic osteomyelitis (CO) and spinocellular carcinoma (SCC): evidence of the main features in differential diagnosis.

	MRI Signal			Diagnosis		
	*T1*	*Stir*	*T1 MoC*	*AO*	*CO*	*SCC*
**Bone Marrow Edema**	Low	High	High (Increased)	Common	Less common ^#^	=CO
**Intraosseous abscess**	Low	High	Peripheral Enhancement	Uncommon	Uncommon ^#^	=CO
**Cortical bone**	Low	Low	No Enhancement	Normal or Periosteal Elevation	Thickened, extensive remodeling	=CO
**Sinus tract**	Low	High	Peripheral Enhancement	**Never**	**Very common**	**Always**
**Sequestrum**	Low	Low	Peripheral Enhancement	Never	Less Common	=CO
**Extraosseous soft tissue changes**	Low	High	Enhancement	**Very** **common**	**Less Common ***	**Mass developing in the epithelial tract of the fistula**

* Limited to the area around sinus tract; ^#^ Common in reactivation.

## Data Availability

Not applicable.

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
