# Peer review of "Management of Squamous Cell Carcinoma in Chronic Osteomyelitis: Our Experience, Review of the Literature and Role of MRI in Differential Diagnosis"

_diagnostics, 2022, doi:10.3390/diagnostics12092062_

Round 1

Reviewer 1 Report

We think that you organized the rare case well as a whole. We recommend editing for several things.

1. For what reason did you say "Multidisciplinary approach" in the title is "multi-"?

2. There is an overlapping expression in lines 15 and 17. Please delete the sentence on line 15.

3. Please describe the criteria by which the authors define CO.

4. In your opinion, at what intervals should MRIs be performed during the follow-up period? Please describe the methods of the authors.

5. With that 6 case, can you conclude the solution for C) "appropriate" presented in the introduction? This requires correction.

Author Response

Thank you for your suggestion; they have been very useful to us.

  1. We used the term “multidisciplinary approach” talking about the collaboration between Orthopedics and Radiologist in management of this long term complication of CO; after your suggestion we change the title removing “multidisciplinary approach” because this collaboration can’t really considered “multi-“

  1. I changed the sentence on line 15

  1. I described the criteria we used to define CO

  1. We routinely use constrast-enhancement MRI during follow-up of patients with long-lasting story of CO and in particular in patients with clinical signs that suggest malignant degeneration of the fistula orifice. After your suggestion, we added the methods we use during follow-up of these patients.

  1. We used the term “correct” not only in relation to our limited experience but also according to the experience that other authors reported in literature.

Reviewer 2 Report

This is an interesting retrospective analysis of cases of Squamous Cell Carcinoma in Chronic Osteomyelitis patients.

I have some questions for the Authors:

- they present a case series of patients with CO since 1995 to 2019; they emphazied the usefulness of gadolinium-enhanced MRI in detecting and Diagnosing tumoral degeneration of soft tissues around CO; the MRI machine they used has been released and commercialized around 2017-2018 so, probably, the number of cases studied with MRI is lower; can the Authors better specify how many patients were studied using the MRI protocol? And can the Authors better specify how many MRI-positive and MRI-negative were in the case series presented?

- did the findings of this study modify the Authors’ clinical protocols in case of CO? In other terms in case of CO patient with more than 20 years of disease do the Authors perform a biopsy or a MRI?

- did the MRI images change the surgical approaches( considering that the amputation is the best treatment in te she cases, especially in the lower limbs)?

Author Response

Thank you for your suggestions; they have been very useful to us.

  1. The COs reviewed were treated between 1995 and 2019, but the SCCs were treated between 2010 and 2019. The MRI machine we used was available in our hospital from 2009; all the patients of this study were studied using that machine. MRI “characteristics” were positive in all the patients with diagnosis of SCC. Unfortunately, we didn’t perform contrast-enhancement MRI in all patients with diagnosis of CO but only in patients with clinical signs suggesting malignant transformation.

  1. In case of long-lasting CO we used MRI not routinely but only in patients with clinical signs suggesting malignant degeneration of fistula orifice. After a “positive” MRI we performed a fine-needle or an incisional biopsy to confirm the diagnosis.

  1. Constrast-enhanced MRI allows to better define surgical margins, helping the orthopedic surgeon to plan the surgical approach between amputation and limb salvage surgery.

According to your last useful suggestions, we modified part of our discussion with the aim of answer to your questions.

Round 2

Reviewer 1 Report

It seems that the points I suggested to the authors are well reflected. Thank you.